# Mechanism of Membrane Fouling Control by HMBR: Effect of Microbial Community on EPS

**DOI:** 10.3390/ijerph17051681

**Published:** 2020-03-05

**Authors:** Qiang Liu, Ying Yao, Delan Xu

**Affiliations:** 1School of Environmental Engineering, Xuzhou University of Technology, No. 2 Lishui Road, Xuzhou 221111, China; xdlxw@126.com; 2Foreign Environmental Cooperation Center, Ministry of Ecology and Environment, No. 5 Houyingfang Hutong, Xicheng District, Beijing 100035, China; yaoying_bfsu@163.com

**Keywords:** hybrid membrane bioreactor (HMBR), membrane fouling, extracellular polymeric substances (EPS), microbial community, biofilm

## Abstract

A hybrid membrane bioreactor (HMBR) employing activated sludge and biofilm simultaneously is proved to represent a good performance on membrane fouling control compared to conventional membrane bioreactor (CMBR) by reducing extracellular polymeric substances (EPS), especially bound EPS (B-EPS). In order to better understand the mechanism of membrane fouling control by the HMBR in regard of microbial community composition, a pilot scale HMBR operated to treat domestic wastewater for six months, and a CMBR operated at the same time as control group. Results showed that HMBR can effectively control membrane fouling. When transmembrane pressure reached 0.1 MPa, the membrane module in the HMBR operated for about 26.7% longer than that in the CMBR. In the HMBR, the quantity of EPS was significantly lower than that in the CMBR. In this paper, soluble EPS was also found to have a close relationship with cake layer resistance. The species richness and diversity in the HMBR were higher than those in the CMBR, and a certain difference between the compositions of microbial communities in the two reactors was confirmed. Therefore, the difference in microbial community compositions may be the direct reason why EPS in the HMBR was lower than that in the CMBR.

## 1. Introduction

In recent years, submerged membrane bioreactors (MBRs) have been more frequently used for wastewater treatment and reclamation. They possess many advantages over conventional activated sludge treatment processes including biostability, high effluent quality, small footprint, and low sludge production rate [1,2,3,4,5]. However, membrane fouling is still the major obstacle preventing the universal application of MBR. Regarding the fouling mechanisms, organic foulants generated from biological process have been the main focus for previous studies, and extracellular polymeric substances (EPS) have been considered the major cause of membrane fouling in MBR [6,7,8,9]. Extracellular polymeric substances are a complex high molecular-weight mixture of polymers excreted by microorganisms. Produced from cell lysis and hydrolysis, they are composed of various organic substances such as carbohydrates, protein, humic substances, uronic acids, and nucleic acid substances [10]. Located at or outside the cell surface, they are often defined as bound EPS (B-EPS) which usually have a dynamic double-layer structure, the outer layer of which is called loosely bound EPS (LB-EPS) and the inner layer tightly bound EPS (TB-EPS) [11,12]. There are also soluble cellular components as the dissolution products of the B-EPS, which are generally called soluble EPS (S-EPS) or soluble microbial products (SMP) [13].

Since EPS are the metabolic products of microorganisms, information about their microbial community compositions, such as microbial community structure, is a prerequisite to fundamental understanding of membrane fouling in the MBR [14,15]. So far, the microbial community in MBR has been investigated by various molecular techniques based on the 16S ribosomal RNA (rRNA) sequences such as fluorescence in situ hybridization (FISH) [16], terminal restriction fragment length polymorphism (T-RFLP) [17] and denaturing gradient gel electrophoresis (DGGE) [18]. Pyrosequencing technology can generate numerous DNA sequences in a single run, providing information on species richness and diversity [19]. High-throughput sequencing (HTS) has made a revolutionary change compared to traditional sequencing technology, with which hundreds of thousands to millions of DNA molecules can be sequenced at one time. Therefore, it is also referred to as the next-generation sequencing (NGS) in some literature.

Many studies explored innovative technologies which can assist membrane fouling control. In the author’s previous studies [20,21], a hybrid membrane bioreactor (HMBR) was developed by introducing biofilm carriers into a conventional membrane bioreactor (CMBR), and it operated to treat municipal wastewater for about one year. With the introduction of biofilm, this novel technology represented good performance on organic and nutrients removals. Furthermore, the increase of transmembrane pressure (TMP) was slowed down and the operating period was prolonged significantly, which was another advantage over the CMBR. In other words, the HMBR represented a good performance on membrane fouling control.

Previous studies [22] showed that HMBR decreased EPS concentration, especially B-EPS, effectively than CMBR. Bound EPS are defined as microbial products, and their effect on the formation of microbial aggregates is important [13]. Since these microbial aggregates are the main component of the activated sludge, their property may have a strong effect on the physical properties of the sludge. In the author’s previous study [23], results showed that B-EPS had a strong correlation with the activated sludge characteristics. Lower B-EPS in the HMBR brought about better flocculability, better settleability, larger floc size, and more compact floc structure [24]. It also resulted in lower cake layer resistance and much slower TMP increase. However, little research has been conducted on the mechanism of membrane fouling control by the HMBR in regard of microbial community composition.

In this paper, a pilot scale HMBR employing activated sludge and biofilm simultaneously in the aeration tank was operated to treat domestic wastewater for approximately half a year, and a CMBR just employing activated sludge was operated at the same time as a control group. The influence of microbial community composition on EPS and their roles on membrane fouling of the HMBR were studied.

## 2. Materials and Methods

### 2.1. Experimental Setup

As shown in Figure 1, the HMBR setup used in this study consisted of an aeration tank equipped with a hollow fiber microfiltration (MF) membrane module, associated with biofilm carriers, pumps, air diffusers, etc. The effective volume of the aeration tank was 1 m^3^. The membrane with the pore size of 0.1 μm was made of enhanced polyvinylidene fluoride (PVDF) and the MF membrane module with the area of 10 m^2^ was manufactured by Hangzhou Kailv Membrane Co. Ltd, (Hangzhou, China). The K_3_ biofilm carriers were made of polyethylene and its volume fraction was controlled at about 50%. The CMBR setup was basically the same as that of the HMBR, except that no biofilm carriers were in the tank.

### 2.2. Raw Wastewater Characteristics

The experiment was conducted in a domestic wastewater treatment plant in Xuzhou, China. The raw wastewater fed to the setup was collected from the sedimentation basin outlet of the plant and its characteristics are shown in Table 1.

### 2.3. Operational Conditions

The hydraulic retention time (HRT) was controlled at 10 h. The sludge retention time (SRT) was controlled at 10 d. The dissolved oxygen (DO) was maintained at about 1.0 mg/L. The membrane flux was maintained at 10 L/m^2^·h. The membrane module performed under intermittent operations of 8 min at 2 min intervals by the suction pump in order to control concentration polarization. When the value of TMP reaches 0.1 MPa, the membrane module should be cleaned according to Huang et al. [25].

### 2.4. Analytical Methods

Samples were collected from the influent, mixed liquor and effluent 2 or 3 times per week. Mixed liquid suspended solids (MLSS) and mixed liquid volatile suspended solids (MLVSS) were analyzed according to gravimetric method. Chemical oxygen demand (COD) was analyzed according to potassium dichromate method. Biological oxygen demand (BOD_5_) was analyzed according to dilution inoculation method. Ammonia nitrogen (NH_4_^+^-N) was analyzed according to Nessler’s reagent spectrophotometry. Total nitrogen (TN) was analyzed according to potassium persulfate oxidation—ultraviolet spectrophotometry. Total phosphorus (TP) was analyzed according to Mo–Sb anti-spectrophotometry [26]. The biofilm biomass was measured according to Liu et al. [21]. EPS were extracted according to Wang et al. [23]. Soluble EPS, LB-EPS and TB-EPS were determined as the sum of carbohydrate and protein per gram MLVSS. Carbohydrate was measured according to anthrone colorimetry [27], and protein was measured according to revised Lowry method [28]. Total membrane resistance (R_t_), membrane intrinsic resistance (R_m_), pore blocking resistance (R_p_), and cake layer resistance (R_c_) were determined following Darcy’s law [29], which can be expressed as:(1)$$J=\frac{\Delta P}{\mu R_{t}}=\frac{\Delta P}{\mu \left(R_{m}+R_{p}+R_{c}\right)}$$
where *J* = membrane flux; Δ*P* = TMP; *μ* = absolute viscosity.

### 2.5. Microbial Community Analysis

The microbial community in the setup was detected by the high-throughput sequencing (HTS) technology. The specific method of HTS was introduced as follows:(1)DNA extraction. The DNA was extracted from the samples using the MoBio PowerSoil DNA extraction kit (MO BIO Laboratories, Carlsbad, CA, USA) following the manufacturer’s instructions.(2)Polymerase chain reaction (PCR) amplification. PCR amplification of 16S rRNA genes was performed using general bacterial primers 515F (5′-GTGCCAGCMGCCGCGGTAA-3′) and 926R (5′-CCGTCAATTCMTTTGAGTTT-3′). The primers also contained the Illumina 5′overhang adapter sequences for two-step amplicon library building.(3)Miseq HTS. The barcoded PCR products were purified using a DNA gel extraction kit (Axygen, China) and quantified using the FTC-3000 TM real-time PCR. The libraries were sequenced by 2 × 300 bp paired-end sequencing on the MiSeq platform using MiSeq v3 Reagent Kit (Illumina, San Diego, CA, USA) at TinyGene Bio-Tech (Shanghai) Co., Ltd., China.(4)Bioinformatic analysis. The raw fastq files were demultiplexed based on the barcode. Paired-end (PE) reads for all samples were run through Trimmomatic (version 0.35) to remove low-quality base pairs. Trimmed reads were then merged using FLASH program (version 1.2.11). The low quality contigs were removed based on screen.seqs command in mothur (version 1.33.3). The cleaned reads were clustered at 97% sequence identity into operational taxonomic units (OTUs) using the UPARSE pipeline (usearch version v8.1.1756). The OTU representative sequences were assigned for taxonomy against Silva 128 database by the classify.seqs command in mothur. Taxonomies (from phylum to species) of the OTUs were determined depending on National Center for Biotechnology Information. Based on the taxonomy, the statistical analysis of community structure was carried out at the level of phylum, class, order, family, genus and species.

## 3. Results and Discussions

### 3.1. Overall Set-Up Performance

During the whole experimental period, MLSS in the CMBR was in the range of 3670–3820 mg/L, and the averaged MLSS was 3740 mg/L. Meanwhile, MLSS in the HMBR was at the same level as that in the CMBR. The biofilm in the HMBR was in the range of 1680–1830 mg/L, so the total averaged biomass in the HMBR was 5490 mg/L. Due to the introduction of biofilm and low DO of about 1.0 mg/L, HMBR represents better performance on organic and nutrients removals compared to CMBR (Table 2). Organic removal is correlated to biomass, which means that high biomass can make high organic removal. Averaged effluent COD of the CMBR was 13.7 mg/L, and the corresponding averaged COD removal was 86.1%. While averaged effluent COD was 9.6 mg/L in the HMBR, corresponding COD removal was 90.2%, higher by 4.1% compared to that in the CMBR. For the same reason, the amount of BOD_5_ removal was higher by 4.2% in the HMBR compared to that in the CMBR. Owing to the low DO, anoxic zone could exist within not only the activated sludge but also the biofilm, and thus TN and TP removals by the HMBR were both enhanced compared to the CMBR.

### 3.2. TMP Variation

Transmembrane pressure (TMP) is linearly related to the membrane filtration resistance. Under the condition of constant flux, the higher the membrane filtration resistance is, the higher the TMP will be. As shown in Figure 2, the rising trend of TMP in the HMBR slowed down noticeably. When TMP reached 0.1 MPa, the membrane module in the HMBR operated for 57 days, while the module in the CMBR operated for only 45 days. The operational period of the membrane module in the HMBR was longer by 26.7% than that in the CMBR, indicating that HMBR has a good performance on membrane fouling control.

On Day 45 when the TMP in the CMBR reached 0.1 MPa, the TMP in the HMBR was only 0.081 MPa, and the membrane resistances in the CMBR and the HMBR in this moment were obtained by calculation. As shown in Table 3, R_c_ was the major component of total membrane resistance, accounting for 75.4% and 69.7% of R_t_ in the CMBR and the HMBR respectively. Moreover, R_p_ and R_c_ in the HMBR were lower by 19.0% and 25.2%, respectively, compared to those in the CMBR, and the removal efficiency of R_c_ by the HMBR was obviously higher than that of R_p_.

### 3.3. EPS Distribution

Mixed liquor samples were taken from the aeration tanks of the CMBR and the HMBR respectively, and various EPS, including S-EPS, LB-EPS, and TB-EPS, were extracted and determined subsequently. Results showed that TB-EPS had the largest share of total EPS and was in the range of 57.8–63.4%; LB-EPS had the second largest share of total EPS and was in the range of 22.3–22.5%; and S-EPS had the least share of total EPS and was in the range of 14.1–14.3% (Figure 3). Apparently, HMBR could reduce EPS effectively, where concentrations of S-EPS, LB-EPS, and TB-EPS were lower by 22.1%, 20.5%, and 0.5% respectively than those in the CMBR. Moreover, in each kind of EPS, carbohydrate was much more than protein. The ratio of carbohydrate to protein in the CMBR was about 2.5, while it was 2.8 in the HMBR, indicating that HMBR reduces protein more efficiently than carbohydrate.

### 3.4. Relationship between EPS and R_c_

Since R_c_ was the major component of R_t_ and the cake layer sludge mainly derived from the activated sludge, it is necessary to study the effect of EPS on the activated sludge characteristics before understanding the relationship between EPS and R_c_. Bound EPS has been proved to have linear relationship with the activated sludge characteristics. When the concentration of B-EPS rose, activated sludge flocculability and settleability deteriorated [22,24]. It was also shown experimentally that as the B-EPS increased, the specific cake resistance became higher [30]. The increasing EPS concentration was found to be one of the factors causing flux decline in the membrane-coupled activated sludge [31].

As for the effect of S-EPS on membrane fouling, most scholars believe that it mainly affected R_p_, because only the substances smaller than the membrane pore can pass through membrane pore and influence R_p_ [32,33,34]. To better understand the effect of S-EPS on membrane fouling, the relationship between S-EPS and R_c_ was studied in this paper. As shown in Figure 4, a good linear relationship was found between S-EPS and Zeta potential. With the decrease of S-EPS, the Zeta potential (absolute value) decreased accordingly. Zeta potential is related to the sludge flocculability, which means the smaller Zeta potential is, the smaller electrostatic repulsion between the colloids will be. Therefore, the sludge flocculability increased when the Zeta potential decreased. As a result, the averaged values of supernatant turbidity in the CMBR and the HMBR were 8.2 NTU and 6.5 NTU respectively, meaning that the sludge flocculability in the HMBR is higher by 20.7% compared to that in the CMBR.

As the cake layer mainly derives from the activated sludge, a close correlation between S-EPS and cake layer specific resistance was found subsequently (see Figure 5). With the decrease of S-EPS, the cake layer specific resistance, i.e., the value of R_c_, decreased correspondingly.

### 3.5. Microbial Community Structure

In order to reveal the reason of EPS difference between CMBR and HMBR, microbial community structures in the CMBR and the HMBR were detected. As shown in Figure 6, both HMBR and CMBR have four major microorganisms, i.e. microorganisms with the highest concentration, namely, *bacteroidetes*, *proteobacteria*, *acidobacteria*, and *saccharibacteria*. The four microorganisms in the HMBR accounted for 44.9%, 29.3%, 8.7%, and 5.3% respectively of all, while it was 40.3%, 27.8%, 13.3%, and 4.3%, respectively, in the CMBR. It is clear that concentrations of *bacteroidetes*, *proteobacteria*, and *saccharibacteria* in the HMBR were higher by 4.6%, 1.5%, and 1.0% respectively, while concentration of *acidobacteria* in the HMBR was lower by 4.6%, compared to those in the CMBR. Higher level of *bacteroidetes* and lower level of *acidobacteria* may be the direct reason why HMBR can reduce EPS effectively compared to CMBR.

### 3.6. Alpha Diversity Analysis

The abundance and diversity of the microbial community can be reflected by single sample diversity analysis (alpha diversity analysis), including a series of statistical analysis indices to estimate the abundance and diversity of species in the environment.

The Alpha diversity includes Chao, Ace, Shannon, and Simpson indices, etc. The larger the first three indices are and the smaller the fourth index is, the richer the species in the sample will be.

Both Chao and Ace indices reflect the species richness in the community, which is the species quantity in the community, regardless of abundance of each species in the community [35]. Moreover, the dilution curves corresponding to the two indices can also reflect whether the sample sequencing amount is sufficient. If the curve tends to be flat or plateau, it shows that the sequencing depth has been covered by all species in the sample substantially. Otherwise, it indicates that the species diversity is too high, and some species cannot be detected.

Both Shannon and Simpson indices reflect the community diversity and are affected by the species richness and evenness in the community [36]. In the case where species richness is at the same level, the greater the species evenness in the community is, the greater the community diversity will be.

In this study, results showed that Chao, Ace, Shannon and Simpson indices in the HMBR were 467.92, 469.71, 3.69 and 0.1264, respectively, while they were 455.77, 446.17, 3.48, and 0.1437, respectively, in the CMBR, indicating that the species richness and diversity in the HMBR were higher than those in the CMBR (Figure 7).

### 3.7. Beta Diversity Analysis

Different from Alpha diversity analysis, Beta diversity analysis is used to compare the difference between a pair of samples in species diversity [37]. It is used to analyze the content of various groups in samples, and then to calculate the Beta diversity value of different samples.

Many indices can represent Beta diversity, such as the commonly used Bray–Curtis, weighted unifracture and unweighted unifrac [38]. Bray–Curtis distance is a common index to reflect the difference between two communities. The calculation of Bray–Curtis distance does not take into account the evolution distance between sequences, but only considers the species existence in the sample. The value of Bray–Curtis distance is between 0–1. The larger the value is, the greater the difference between the samples will be.

Beta diversity matrix heatmap, by visualizing the Beta diversity data by graphics and clustering together the samples with similar Beta diversity, reflects the similarity between samples.

In this paper, the difference between microbial communities of activated sludge in the CMBR and the HMBR was calculated by Bray–Curtis at phylum level, and results showed that the Bray–Curtis distance at phylum levels was 0.0883, indicating that sludge community in the HMBR had slight differences with that in the CMBR. 

### 3.8. Possible Mechanism of Membrane Fouling Control by HMBR

With the biofilm addition, the microbial community composition in HMBR showed a clear difference with that in CMBR. As shown in the results mentioned in Section 3.5, although types of microorganisms with the highest concentration in the HMBR were the same as those in the CMBR, including *bacteroidetes*, *proteobacteria*, *acidobacteria*, and *saccharibacteria*, concentrations of these four microbes in the CMBR and the HMBR were obviously different, with higher concentrations of *bacteroidetes*, *proteobacteria*, and *saccharibacteria*, and lower concentration of *acidobacteria* in the HMBR. Furthermore, both the species richness and diversity in the HMBR were apparently higher than those in the CMBR, and a certain difference between compositions of microbial communities in the two reactors was confirmed. From above, it can be concluded that the difference in microbial community composition was the possible direct reason for the EPS difference between CMBR and HMBR.

Hybrid membrane bioreactor can effectively reduce EPS concentration, including S-EPS, LB-EPS and TB-EPS [22]. In this paper, concentrations of the three components in the HMBR were lower by 22.1%, 20.5% and 0.5% respectively compared to those in the CMBR. The decrease of B-EPS has been proved to improve flocculability and settleability of activated sludge and decrease R_c_ [29,30,31]. Soluble EPS is generally believed to be related with R_p_, because only the substances smaller than the membrane pore could pass through the membrane [32,33,34]. In this paper, the relationship between S-EPS and R_c_ was studied, and results showed that S-EPS has a linear correlation with Zeta potential. As S-EPS decreases, lower Zeta potential (absolute value) is obtained. Since Zeta potential is related to the sludge flocculability, the lower Zeta potential is, the smaller the repulsion between the colloids will be, and the better the sludge flocculability will be found. As the cake layer sludge mainly derives from the activated sludge, improvement of activated sludge flocculability decrease the cake layer resistance. Therefore, a close relationship is established between S-EPS and the cake layer specific resistance. With the decrease of S-EPS, the cake layer specific resistance decreases accordingly. In addition, there may also be another way by which EPS can affect the value of R_c_. The cake layer adheres to the membrane surface and forms a secondary membrane. During the filtration process, S-EPS must first pass through the cake layer pore before getting through the membrane pore, so the S-EPS decrease would reduce the resistance of cake layer pore. As a result, with the S-EPS and B-EPS decrease, HMBR represents a good behaviour on membrane fouling control. Its membrane module operational period was prolonged by 26.7% compared to the CMBR. On the other hand, when CMBR and HMBR operated for the same period of time, 45 days in this case, concentrations of R_p_ and R_c_ in the HMBR were lower by 19.0% and 25.2% respectively compared to those in the CMBR.

## 4. Conclusions

In this study, particular attention has been paid on the mechanism of membrane fouling control by HMBR in regard of microbial community composition. The following conclusions are drawn:
Hybrid membrane bioreactor can remove organics and nutrients effectively. Averaged COD, BOD_5_, NH_4_^+^-N, TN and TP removals by which were higher by 4.1%, 4.2%, 1.0%, 17.7% and 2.2% respectively compared to those in the CMBR. Moreover, HMBR represents a good performance on membrane fouling control. When TMP reached 0.1 MPa, the membrane module in the HMBR operated for 57 days, which was longer by 26.7% compared to that in the CMBR.Soluble EPS in the HMBR was lower by 22.1% compared to the CMBR. Due to the S-EPS decrease, Zeta potential decreased, with activated sludge flocculability increasing and cake layer specific resistance decreasing accordingly. As a result, when the two reactors operated for the same time length of 45 days, R_p_ and R_c_ in the HMBR were lower by 19.0% and 25.2% respectively compared to those in the CMBR.A certain difference, albeit small, between the microbial community structures in the CMBR and HMBR was confirmed. Besides, both the species richness and diversity in the HMBR was apparently higher than those in the CMBR. This may be the direct reason why the HMBR can reduce the EPS effectively compared to the CMBR.

## Figures and Tables

**Figure 1 ijerph-17-01681-f001:**
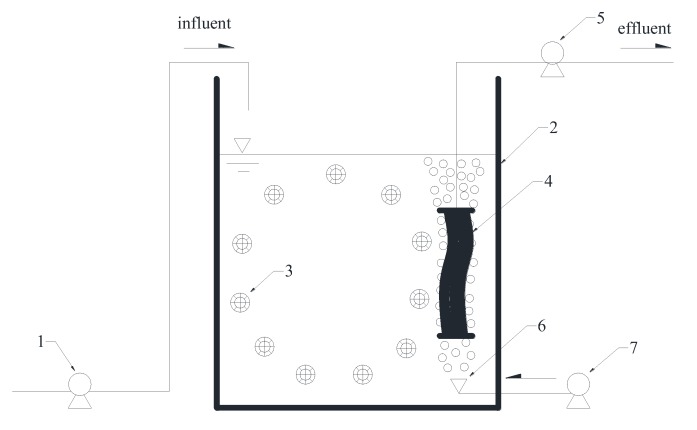
Schematic diagram of the hybrid membrane bioreactor (HMBR) setup. 1. feed pump; 2. aeration tank; 3. biofilm carrier; 4. microfiltration membrane module; 5. suction pump; 6. air diffuser; 7. air compressor.

**Figure 2 ijerph-17-01681-f002:**
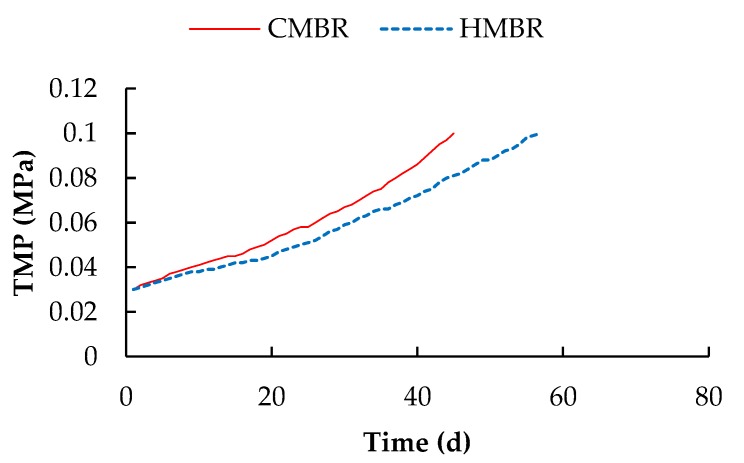
Transmembrane pressure (TMP) Variation in conventional membrane bioreactor (CMBR) and hybrid membrane bioreactor (HMBR).

**Figure 3 ijerph-17-01681-f003:**
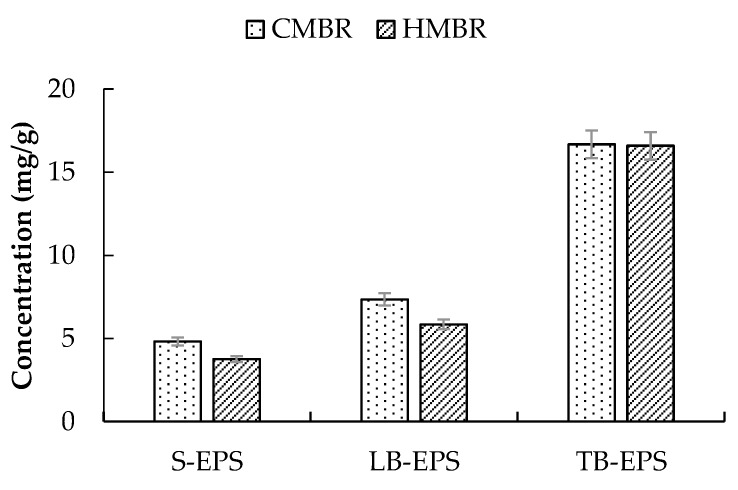
EPS distribution in the conventional membrane bioreactor (CMBR) and hybrid membrane bioreactor (HMBR).

**Figure 4 ijerph-17-01681-f004:**
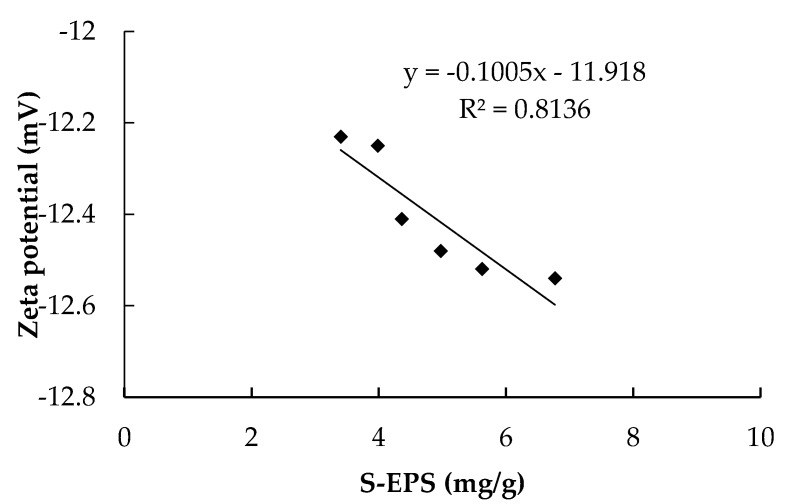
Relationship between soluble extracellular polymeric substances (S-EPS) and Zeta potential.

**Figure 5 ijerph-17-01681-f005:**
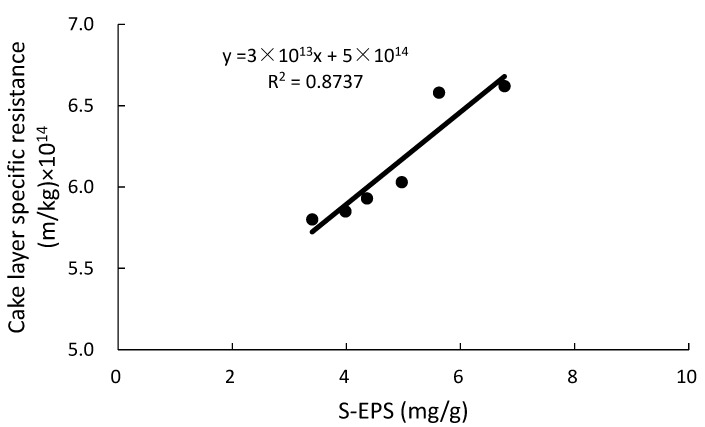
Relationship between soluble extracellular polymeric substances (S-EPS) and cake layer specific resistance.

**Figure 6 ijerph-17-01681-f006:**
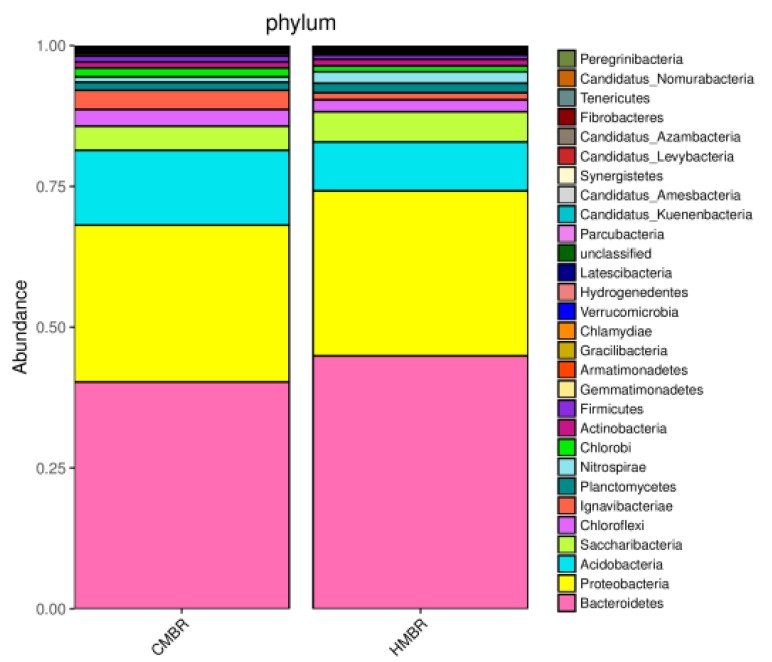
The microbial community structures of activated sludge in CMBR and HMBR at phylum level.

**Figure 7 ijerph-17-01681-f007:**
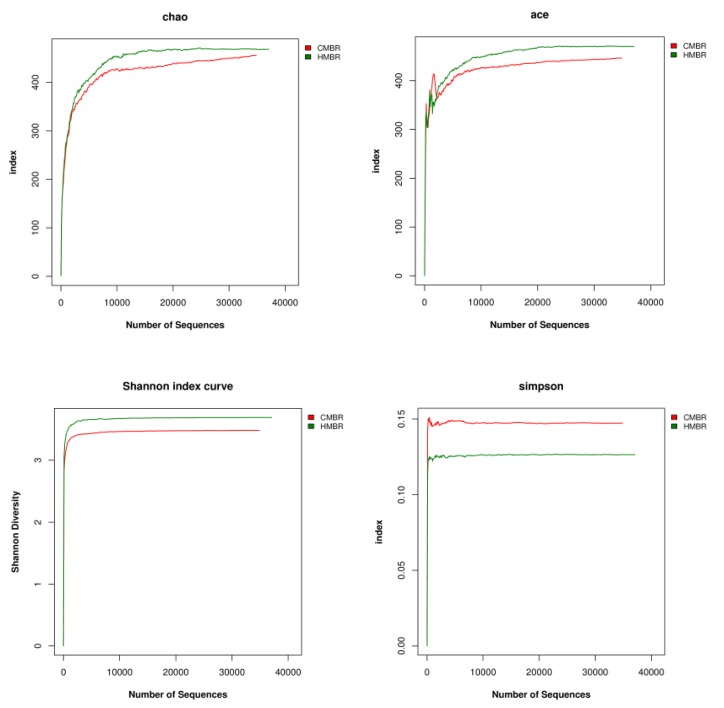
Several index curves about microbial community in CMBR and HMBR.

**Table 1 ijerph-17-01681-t001:** Characteristics of the raw wastewater.

Parameter	Description	Average
Chemical oxygen demand (COD) (mg/L)	76.3–113.6	98.6
Biological oxygen demand (BOD_5_) (mg/L)	45.7–65.2	52.7
Ammonia nitrogen (NH_4_^+^-N) (mg/L)	26.3–34.8	29.2
Total nitrogen (TN) (mg/L)	31.4–52.6	37.4
Total phosphorus (TP) (mg/L)	2.47–3.74	3.20
Temperature (°C)	17.8–22.6	20.3
pH	7.53–7.69	7.61

**Table 2 ijerph-17-01681-t002:** Performance of conventional membrane bioreactor (CMBR) and hybrid membrane bioreactor (HMBR) on organic and nutrients removals.

Parameter	CMBR		HMBR	
	Effluent (mg/L)	Averaged Removal (%)	Effluent (mg/L)	Averaged Removal (%)
COD	9.7–16.3 (13.7)	86.1	6.7–13.1 (9.6)	90.2
BOD_5_	6.5–10.8 (8.9)	83.1	4.8–9.4 (6.3)	87.3
NH_4_^+^-N	0.3–1.1 (0.7)	97.6	0.3–0.8 (0.4)	98.6
TN	21.1–31.7 (25.8)	31.0	17.2–25.1 (19.2)	48.7
TP	0.53–1.06 (0.72)	77.5	0.37–1.12 (0.65)	79.7

Note: Values in brackets are the average.

**Table 3 ijerph-17-01681-t003:** Comparison of membrane resistance for conventional membrane bioreactor (CMBR) and hybrid membrane bioreactor (HMBR).

Operation Mode	Total Membrane Resistance (R_t_) (/m)	Membrane Intrinsic Resistance (R_m_) (/m)	Pore Blocking Resistance (R_p_) (/m)	Cake Layer Resistance (R_c_) (/m)
CMBR	3.58 × 10^13^	0.71 × 10^13^	0.21 × 10^13^	2.70 × 10^13^
HMBR	2.90 × 10^13^	0.71 × 10^13^	0.17 × 10^13^	2.02 × 10^13^

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
