# Peer review of "Mechanism of Membrane Fouling Control by HMBR: Effect of Microbial Community on EPS"

_ijerph, 2020, doi:10.3390/ijerph17051681_

Round 1
Reviewer 1 Report
General comment:
In principle the objective of the study is interesting and relevant (membrane fouling in two different types of a membrane bioreactor). The authors have investigated the processes of membrane fouling in a conventional membrane bioreactor (CMBR) and a hybrid membrane bioreactor (HMBR) with particular regard to the microbial community structure and the different types of extracellular polymeric substances (EPS). The manuscript goes on to evaluate the possible use of the HMBR for the control of membrane fouling.
The following issues should be added/discussed:
- Page 2, lines 72 – 74: The authors describe the effects of lower B-EPS concentrations. Are there any reasons known (or mentioned in literature) why B-EPS causes these effects?
- Page 3, lines 105 - 112: The authors have listed the used analytical methods, but in some cases they should add basic details (e.g. they only mention the “standard methods”). Which analytical method was used for the determination of MLVSS, COD, BOD5, NH4-N+, TN and TP? What are the principles (analytical method) of the “Lowry method” and “determination following Darcy’s law”?
- All personal names should be written with large initial characters, e.g.: Page 2, line 110: Lowry; Page 7/8, lines 221, 223, 229, 233, 244, 245, 248 and 253: Chao, Ace, Shannon, Simpson, Bray-Curtis
- Page 5, Figure 2: The authors should use lines (with different line types) instead of symbols for the two data series.
- Page 7, Figure 6: The labelling (axes and legend) is too small and illegible.
- Page 7, Figure 7: The graphs and the labelling (axes and legend) are too small and illegible.
- Page 9, Figure 8: The figure does not provide any information in the current form (the colour difference is not visible).
- Page 1, lines 25 -27; Page 7, 209 – 211; Page 9, lines 263 – 265; Page 10, lines 312 – 314: Fourfold repetition of the same content (with identical wording).
- In some cases (e.g. concerning microbial community structure) the results for CMBR and HMBR show relatively small differences. Where the results confirmed by a repetition of the experiment or by comparable results from other publications?
In the final analysis, the current version of the paper could be suitable for publication after revision of the entire manuscript and clarification of the outstanding issues.
The specific comments are summarized in the attached pdf file “Specific comments_ijerph-725293”.

Reviewer 2 Report
I have read throughout the manuscript; the overall presentation of the m/s is satisfactory. However, the manuscript should be revised mainly due to the fact that in several instances, the presentation and the discussion of the achieved results and the comparison with the relevant literature are rather limited. I think for this reason the article becomes of little interest.
- The abstract should not be a repetition of conclusion, it should state briefly the purpose of the research, the principal results and major conclusions
- Data presented in table 1 doesn’t have SD value.
- Line no 186 to 188 should be discussed with relevant literature.
- Why only phylum level of microbial community was discussed after molecular analysis of microorganisms?
- α and β diversity should be discussed with proper literature.
- Data were repeated in section 3.8 without any literature support.
- The main conclusions drawn from results should be presented in a short, shouldn't be the repetition of the abstract.
Reviewer 3 Report
This is a very interesting, well-written manuscript, presenting a novel approach to wastewater treatment combining conventional MBR with biofilm carriers. The authors clearly presented the main idea and the aim of the study. MEthods are chosen properly. The presentation of the results is clear and consistent. I recommend improving slightly the discussion, by adding some practical contexts of the implementation in the technical scale, including the determination of the Technology Readiness Level (TRL).
I recommend this manuscript for publication after some minor, technical improvement:
Table 2. If there are results of concentrations in the averaged effluent, the standard deviation values should be provided (to present the scattering of the results).
Figures 4 and 5 - the equations describing the dependences should be given. The determination coefficient is not enough.
Figures 6 and 7 - fonts should be larger. Currently, it is hard to read.
